# Clusters of cause specific neonatal mortality and its association with per capita gross domestic product: A structured spatial analytical approach

**Daniela Testoni Costa-Nobre**[ORCID][1]*, **Mandira Daripa Kawakami**[ORCID][1], **Kelsy Catherina Nema Areco**[1], **Adriana Sanudo**[1], **Rita Cassia Xavier Balda**[1], **Ana Sílvia Scavacini Marinonio**[1], **Milton Harumi Miyoshi**[1], **Tulio Konstantyner**[ORCID][1], **Paulo Bandiera-Paiva**[1], **Rosa Maria Vieira Freitas**[2], **Liliam Cristina Correia Morais**[2], **Mônica La Porte Teixeira**[2], **Bernadette Waldvogel**[2], **Maria Fernanda Branco de Almeida**[1], **Ruth Guinsburg**[1], **Carlos Roberto Veiga Kiffer**[ORCID][1]

**1** Escola Paulista de Medicina–Universidade Federal de São Paulo (UNIFESP), São Paulo, SP, Brazil,
**2** Fundação Sistema Estadual de Análise de Dados (SEADE Foundation), São Paulo, SP, Brazil

* danielatestoni@gmail.com

**Data Availability Statement:** The database of the study was uploaded in the ZENODO public

## Abstract

### Background

Infant mortality rate is a measure of population health and neonatal mortality account for great proportion of these deaths. Underdevelopment might be associated to higher neonatal mortality risk due to assistant related factors. Spatial and temporal distribution of mortality help identifying and developing strategies for interventions.

### Objective

To investigate the cluster areas of asphyxia-associated neonatal mortality and to explore its association with per capita gross domestic product (GDP) in São Paulo State (SP), Brazil.

### Methods

Ecological study including live births residents in SP from 2004–2013. Neonatal deaths (0–27 days) with perinatal asphyxia were defined as intrauterine hypoxia, birth asphyxia or meconium aspiration syndrome written in any line of the Death Certificate. Geoprocessing analytical approach included detection of first order effects through quintiles and spatial moving average maps, followed by second order effects by global and local spatial autocorrelation (Moran and LISA, respectively) before and after smoothing with local Bayesian estimates. Finally, Spearman correlation was applied between asphyxia-associated neonatal mortality and mean per capita GDP rates for the municipalities with significant LISA.

### Results

There were 6,713 asphyxia-associated neonatal deaths among 5,949,267 live births (rate: 1.13/1000) in SP. Spatial moving average maps showed a non-random distribution among

repository. The DOI for the data is: 10.5281/
zenodo.4915807.

**Funding:** Fundação de Amparo à Pesquisa de São Paulo - FAPESP, Project # 2017/03748- 7.

**Competing interests:** The authors have declared that no competing interests exist.

**Abbreviations:** SEADE, Fundação Sistema Estadual de Análise de Dados; GDP, Gross Domestic Product; GIS, Geographic Information System; ICD, International Classification of Diseases; HDI, Human Development Index; LISA, Local Indicators of Spatial Association.

municipalities, with presence of clusters ($I = 0.048$; $p = 0.023$). LISA map identified clusters of asphyxia-associated neonatal mortality in the south, southeast and northwest. After applying local Bayes estimates, clusters were more pronounced ($I = 0.589$; $p = 0.001$). There was a partial overlap of the areas of higher asphyxia-associated neonatal mortality and lower mean per capita GDP.

## Conclusions

Spatial analysis identified cluster areas of high asphyxia-associated neonatal mortality and low per capita GDP rates, with a significant negative correlation. This optimized, structured, and hierarchical approach to identify high-risk areas of cause-specific neonatal mortality may be helpful for guiding public health efforts to decrease neonatal mortality.

## Introduction

Infant mortality rate is a measure of population health [1]. Although there has been a substantial progress in reducing infant mortality in the last 20 years, post neonatal mortality is declining faster than neonatal mortality. Globaly, mortality rates in the first month of life fell by 41% in the last 15–20 years (31/1000 live births in 2000 to 18/1000 live births in 2017), whereas the redution in mortality for children aged 1–59 months was 54% in the same period [2]. In 2017, 2.5 million deaths occurred in children aged under 1 month caused, in most cases, by preterm birth, perinatal asphyxia, infections and birth defects [2]. Most newborn deaths take place in low and middle-income countries [2].

Brazil achieved the 4th *Millennium Development Goal* reducing child mortality by two thirds between 1990 and 2015, and the neonatal mortality also showed a significant reduction from 25/1000 live births in 1990 to 9/1000 live births in 2017 [3]. The 2019 estimated rate was 8.13/1000 live births [4]. Despite its considerable decline over the years, neonatal mortality remains high, with almost 10% of the underlying cause of deaths being classified as asphyxia/ hypoxia [5]. Asphyxia/hypoxia is the 5th cause of neonatal mortality in Brazil following prematurity (30%), congenital malformation (23%), infections (19%) and maternal factors, but it represents one of a pottentialy preventable one [5].

Lack of quality care at birth, or skilled care and treatment immediately after birth and in the first days of life are related with the most common causes of neonatal deaths [4]. Several studies demonstrate the association of infant mortality with social economic factors [6–8]. However, few studies evaluated economic indicators associated with neonatal mortality [9, 10]. Negative economic growth measured by a decrease in the per capita gross domestic product (GDP) (data obtained for 214 countries from 1981 to 2010) was associated with deterioration in several child mortality measures including neonatal, post-neonatal, child and under 5 years of age mortality rates [9]. The impact of a 1% increase in per capita GDP was associated with almost 7% of reduction in neonatal mortality (95% CI -0.09 to -0.05) [9]. Increase in public expenditure on health as a proportion of GDP was associated with a significant reduction in infant mortality rate in Brazil, however only the post-neonatal mortality was associated with increasing public expenditure on health as a proportion of GDP [8]. For Volpe et al., the increase in public expenditure on health reduced the post-neonatal mortality with no impact on neonatal mortality because the strategies for reducing neonatal mortality are more

expensive than those for reducing post-neonatal mortality, for which low-cost public interventions may have a great effect [8].

The use of Geographic Information Systems (GIS) in public health relies on two founding principles: populations are inextricably bound to locations and closer populations have closer relationship than the ones that are more distant from each other [11, 12]. Thus, GIS and spatial analysis may be helpful in explaining neonatal mortality rates and causes over a given area. Spatial analysis may be used to explore the co-occurrence of a health event and other population events on the same geographic unit (i.e. economic, educational, sociological indicators) [13].

Since asphyxia/hypoxia is a leading cause of pottentialy preventable neonatal deaths in Brazil, a methodology to identify high risk areas for these deaths and to explore the co-occurrence of associated events would be helpful for guiding public health strategies. Therefore, our aim was to investigate by a structured and hierarchical spatial analytical method the occurrence of clustering areas of asphyxia-associated neonatal mortality and to explore its association with mean per capita GDP, in São Paulo State, Brazil.

## Methods

### Study design and population

A population-based study applying spatial analysis per area was performed, including all live births from mothers living in São Paulo State from 2004 to 2013. São Paulo State is the richest State of Brazil with a Human Development Index (HDI) of 0.826. It has 645 Municipalities with HDI varying from 0.639 to 0.862 [14]. Exclusion criteria were infants with birth weight < 500g and/or gestational age < 22 weeks, infants with unknown birth weight and gestational age, and infants with congenital anomalies.

The study was approved by the Ethics Committee on Human Research of Escola Paulista de Medicina–Universidade Federal de São Paulo (# 2,580,929) and by the Board of Directors of Fundação SEADE. Informed consent was waived given the retrospective nature of the study and use of unidentified data. The dataset was accessed from October 2018 to June 2021, and it was fully anonymized before the access.

### Definitions

Asphyxia-associated neonatal mortality was defined as any death before 28 days after birth with hypoxia, asphyxia, or meconium aspiration as cause of death in any line (Ia, Ib, Ic or II) of the Death Certificate according to the following diagnosis in the International Classification of Disease (ICD 10, WHO) (Table 1) [15].

Asphyxia-associated neonatal mortality rate per geographic unit was calculated by dividing the total number of asphyxia-associated neonatal deaths by the total number of live births

**Table 1. ICD codes used to define asphyxia-associated neonatal mortality.**

| | |
|---|---|
| P20.0 | Intrauterine hypoxia first noted before onset of labor |
| P20.1 | Intrauterine hypoxia first noted during labor and delivery |
| P20.9 | Intrauterine hypoxia, unspecified |
| P21.0 | Severe birth asphyxia |
| P21.1 | Mild or moderate birth asphyxia |
| P21.9 | Birth asphyxia, unspecified |
| P24.0 | Neonatal meconium aspiration |

from mothers living in that area per 1000 occurrences for each one of the 645 São Paulo municipalities.

The annual per capita GDP in Brazilian Reais (R$) and annual estimated population between 2004 and 2013 were obtained from the public database of *Fundação Sistema Estadual de Análise de Dados* (SEADE Foundation) [16]. The mean per capita GDP for the period was calculated for each municipality by dividing the GDP of each municipality by the estimated population in the same period. We converted the obtained values to US dollars using the conversion rate of the end of the period (1 US$ = 2.34 R$, December of 2013).

## Database

Data was obtained from SEADE Foundation, which extracted each registry from the Live Birth Certificates and the Death Certificates. The database for the deaths was created by deterministic linkage of Death and Live Birth Certificates made by SEADE Foundation analyzing the files of all infants who died until 365 days with the respective registry of live births, with the discordant cases being reviewed. Data management was performed using STATA software (version 15, Stata Corp LP, College Station, TX, USA). The number of deaths and live births for each municipality was added and the crude rate of asphyxia-associated neonatal mortality calculated for each of 645 São Paulo municipalities. The final database was exported to an Access file (.mdb) and contained one registry per municipality with asphyxia-associated neonatal mortality and mean per capita GDP rates.

Geographic coordinates and spatial files of the São Paulo state and respective municipalities in shape file format (.shp) were obtained from the Brazilian Institute of Geography and Statistics (IBGE) [17]. In short, the complete state with each municipality and its respective unique identifiers composed a layer in the geographical database. External tables containing both asphyxia-associated neonatal rate and mean per capita GDP in.mdb format per municipality were imported using the municipality unique identifier to link with shape layer in the geographical database using the TerraView software version 4.2.2 (Instituto Nacional de Pesquisas Espaciais, São José dos Campos, Brazil). A single layer within the database containing both the map files (.shp) and their respective municipality imported tables was created. From this layer, different views were generated for exploring both asphyxia-associated neonatal mortality and mean per capita GDP per municipality, as described below.

## Statistical and spatial analysis

Initially, visualization of the crude rates of asphyxia-associated neonatal mortality and mean per capita GDP per municipality were performed by using theme maps with quintiles distribution. Then, first order effects were explored for each rate (asphyxia-associated neonatal mortality and mean per capita GDP) per municipality using spatial moving average ($\mu_i$) calculated for each municipality based upon rates associated to neighboring municipality, used to detect tendencies and spatial patterns [18].

Second order effects of local spatial dependency and autocorrelation were explored for each rate (asphyxia-associated neonatal mortality and mean per capita GDP) per municipality by applying global Moran indicators (*I*) and Local Indicators of Spatial Association (LISA), which generated LISA cluster (or Moran) maps [18]. LISA cluster (or Moran) map shows municipalities with significant G (significance at 95%), box-plotted according to their dispersion diagram (i.e. areas (Q1) and (Q2) with positive spatial correlation, high-high values and low-low values respectively; areas (Q3) and (Q4) with negative spatial correlation, high-low values and low-high values, respectively). If second order effects suggested spatial autocorrelation, a surface smoothing technique was applied using local empiric Bayesian estimator equation, given by:

$b_i = w_i t_i + (1 - w_i) m_i$, where $w_i$ = weight between 0 and 1, which depends on the size of the population in area I, $t_i$ = risk rate in the area, $m_i$ is the overall rate for the local area [19]. The larger the population in area *I*, the closer weight $w_i$, which means that in areas where the population is larger, the empirical Bayes estimate is very close to the risk rate for that area ($t_i$), and in areas with a small population, the value of $b_i$ will be intermediary between $t_i$ and $m_i$.

The equation was applied on each rate (asphyxia-associated neonatal mortality and mean per capita GDP) per municipality, mean rates were calculated for each municipality according to each of its neighbor areas for reducing the impact of outliers with low occurrences. Robustness testing of global and local spatial autocorrelations were applied for cluster confirmation, after smoothing with local empiric Bayesian estimator.

Finally, the correlation of asphyxia-associated neonatal mortality and mean per capita GDP per municipality rates was qualitatively and quantitatively investigated. Qualitative exploration was performed by visual comparison of overlapping areas in smoothed maps of asphyxia-associated neonatal mortality and mean per capita GDP rates per municipality. Quantitative comparison was performed by non-parametric Spearman correlation test applied between the asphyxia-associated neonatal mortality and the mean per capita GDP rates per municipality, only for the municipalities with significant G in LISA significance maps, as described above. A p-value <0.05 was considered as statistically significant for all analyses.

## Results

From 2004 to 2013, there were 6,090,532 live births and 8,725 neonatal asphyxia-associated deaths in São Paulo State (Fig 1). From those, 2.3% of the live births and 23% of the neonatal asphyxia-associated deaths met the exclusion criteria, resulting in 5,949,267 live births and 6,713 deaths. Mother´s residence municipality was unknown despite being in São Paulo State for 175 live births and 2 deaths. The mean birth weight of the neonates that died with asphyxia was 1,907g (±SD 1,154g); with 17 infants with unknown birth weight. More than half were male (56% [3752/6713]. Regarding gestational age, (37% [2,484/6,642] had 37 to 41 weeks, 15% [978/6,642] 32 to 36 weeks, 16% [1,036/6,642] 28 to 31 weeks, 32% [2,090/6,642] 22 to 27 weeks, and 1% [54/6,642] ≥42 weeks, with 71 infants with unknown gestational age. Maternal age was < 20 years for 21% [1,391/6,713], 20–34 years for 65% [4,361/6,713], and >34 years for 14% [961/6,713]. The asphyxia-associated neonatal mortality rate for the State during the study period was 1.13/1000 live births, varying from 1.37/1000 live births in 2004 to 0.93/1000 live births in 2013. The asphyxia-associated neonatal rate calculated for each of 645 São Paulo municipalities had a median of 1.05/1000 live births (minimum-maximum: 0.00–11.49). The mean per capita GDP for the period varied from US$ 2,048.00 to US$ 58,688.00 among the 645 municipalities.

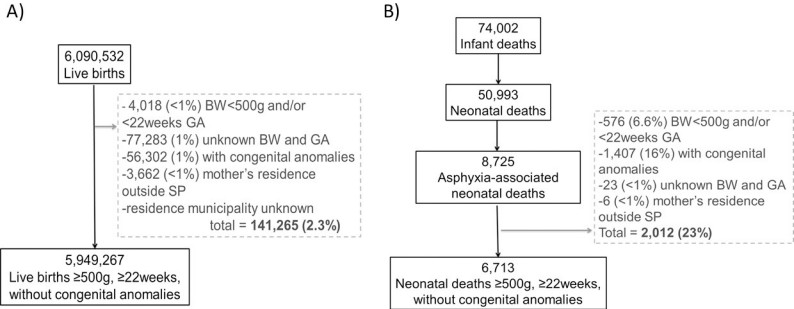

**Fig 1. Flowchart of included population–São Paulo State (Brazil), from 2004 to 2013.** A) Live births; B) Deaths. BW: birth weight, GA: gestational age, SP: São Paulo State.

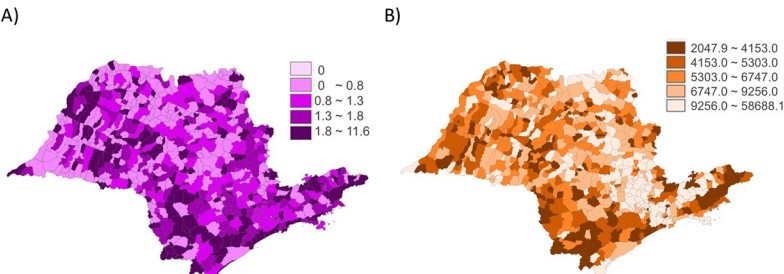

**Fig 2. Neonatal asphyxia-associated mortality and mean per capita GDP crude rates.** A) Raw rates of neonatal asphyxia-associated mortality classified by quintiles. Darker colors show municipalities with higher asphyxia-associated mortality rates. B) Raw rates of mean per capita GDP in US$, classified by quintiles. Darker colors show municipalities with lower mean per capita GDP. Data are presented for each 645 São Paulo Municipalities (Brazil), from 2004 to 2013.

Descriptive maps by quintiles of crude rates of the asphyxia-associated neonatal mortality and the mean per capita GDP rates per municipality suggested that spatial dependence was present, with possible cluster areas of higher and lower mortality and higher and lower per capita GDP (Fig 2).

In addition, descriptive first order maps by quintiles (Fig 3A) and spatial moving average ($\mu_i$) (Fig 3B) of asphyxia-associated neonatal mortality crude rates reinforced the suggestion of the presence of cluster areas. Second order effect showed a weak but significant positive global index result ($I = 0.048$, $p = 0.023$), suggesting a possible non-random spatial distribution. After applying local Bayesian estimation, clusters were more evident (Fig 3C and 3D), confirming areas in the South and Southeast with high mortality rates and areas in the Northwest with low mortality rates. After smoothing, global Moran index revealed a stronger positive and significant overall spatial autocorrelation ($I = 0.589$, $p = 0.001$).

The same approach for mean per capita GDP rates per municipality before smoothing showed a first order effect of potential cluster areas of low mean per capita GDP rates. Second order global Moran index was positive and significant ($I = 0.272$, $p = 0.001$), indicating a non-

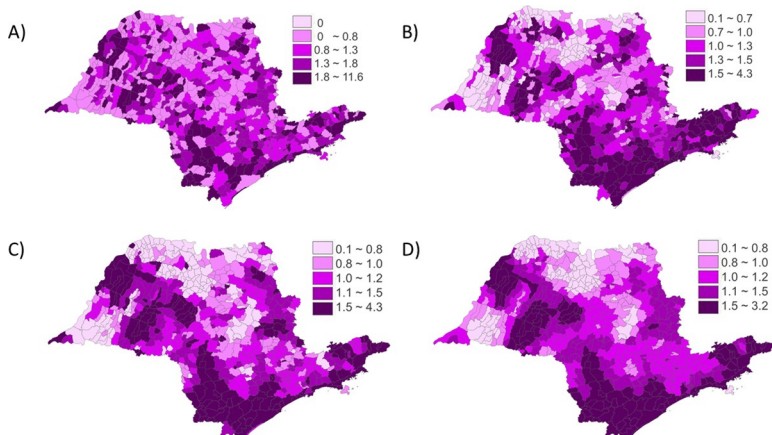

**Fig 3. Neonatal asphyxia-associated mortality rates distributed by quintiles and spatial moving average ($\mu_i$) before and after smoothing.** A) Rates by quintiles before smoothing with local Bayes estimates. B) Rates by $\mu_i$ before smoothing with local Bayes estimates. C) Rates by quintiles after smoothing with local Bayes estimates. D) Rates by $\mu_i$ after smoothing with local Bayes estimates. Data are presented for each 645 São Paulo Municipalities (Brazil), from 2004 to 2013.

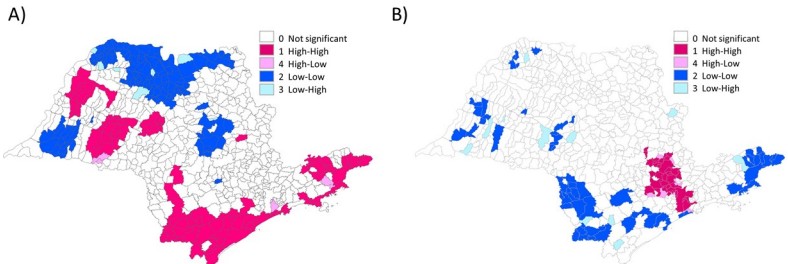

**Fig 4. Lisa cluster map of neonatal asphyxia-associated mortality and mean per capita GDP rates from 2004 to 2013, in São Paulo State (BR).** A) Neonatal asphyxia-associated mortality after smoothing with local Bayes estimates. B) Mean per capita GPD. Maps (A) and (B) represents municipalities with significant G box-plotted according to their dispersion diagram, i.e. areas (1) and (2) with positive spatial correlation, and areas (3) and (4) with negative spatial correlation. Data are presented for each 645 São Paulo Municipalities (Brazil), from 2004 to 2013.

random distribution. Smoothing by local Bayesian estimation did not significantly alter the mean per capita GDP rates per municipality, with no impact on global Moran index ($I$ = 0.272, $p$ = 0.001).

Clusters in LISA cluster maps of asphyxia-associated neonatal mortality were easier to visualize after smoothing with local Bayesian estimation and the number of municipalities with significant spatial autocorrelation with neighboring areas was higher than before. LISA cluster map identified potential clusters of asphyxia-associated neonatal mortality in the South, Southeast and Northwest of the State (Fig 4A). Since smoothing approach did not alter the results for the mean per capita GDP rates, LISA cluster maps for the mean per capita GDP rates was depicted with no smoothing technique, while maps after smoothing were depicted for neonatal asphyxia-associated mortality rates. There were potential cluster areas of low mean per capita GDP rates in the South and Southeast of the State and a belt of possible cluster areas of higher mean per capita GDP rates in North and the Southeast areas of the State (Fig 4B).

A total of 45 municipalities had a significant G in the LISA significance map for both asphyxia-associated neonatal mortality after Bayesian smoothing and mean per capita GDP rates. Spearman's rank correlation coefficient showed a significantly dissimilar (opposed) result ($r_s$ = -0.59, 95% CI -0.76 to -0.36, p<0.001) and a scatter diagram with lognormal fit curve containing these 45 municipalities is shown in Fig 5. There were 63 municipalities in

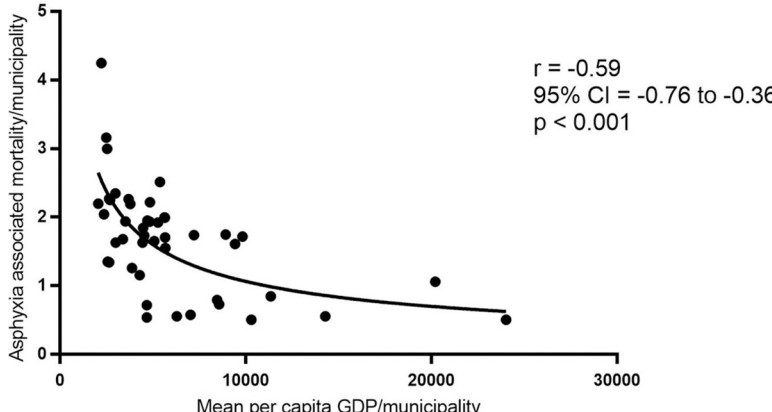

**Fig 5. Dispersion of mean per capita GDP rates and asphyxia-associated mortality rates per municipality.** Data presented for 45 municipalities with significant G in LISA significance map with lognormal fit curve and Spearman correlation (r), São Paulo (Brazil), from 2004 to 2013.

clusters of high asphyxia-associated neonatal deaths but no significance (by LISA) for mean per capita GDP rates.

## Discussion

We found a mean asphyxia-associated neonatal mortality rate of 1.13 per 1000 live births (SD ± 1.29) for São Paulo municipalities, in infants ≥ 500g, ≥ 22 weeks gestation, without congenital anomalies, born from mothers living in the State. Clusters of asphyxia-associated neonatal mortality were identified in the South, Southeast and Northwest of the State. Clusters of low mean per capita GDP rates were shown in South and Southeast of the state. There was a negative significant correlation between asphyxia-associated neonatal mortality after Bayesian smoothing and mean per capita GDP rates.

In 2015, the estimated asphyxia mortality rate in Brazil was 1.9 per 1000 live births and 1.5 per 1000 live births in São Paulo State, according to the Global Burden of Disease Study (GDB 2015) [20]. The GDB study estimated deaths using several sources of data, such as the Mortality Information System (SIM), the demographic census, the National Household Sample Survey (PNAD), and the Demographic and Health Survey (PNDS) [20]. The present study used civil registry data from death certificates to collect the cause of death. Estimation of deaths are useful for inferring rates but, to define the leading causes of child mortality, the above cited study used a list to sort the causes of death, and death reassignment for unspecific causes of death. Meanwhile, census or universal coverage (national or state registries) data represent the actual health phenomena but may be impacted by the reliability of the underlying cause of death. Underlying cause of death in the death certificate is uncertain, since it is an information based on the physician's perception and coded using the International Classification of Diseases (ICD-10). A previous study compared the information of death certificates from 452 infants with hospital records and found a poor agreement (only 38%) between the originally declared and the modified underlying cause of death defined by the authors according to the patient medical record [21]. Therefore, since the underlying cause of death in the database could be misleading, a strategy for improving the reliability of asphyxia-associated mortality data of the present study was to consider the term "asphyxia" in all lines of the death certificate, analyzing all deaths associated to asphyxia.

We identified the spatial distribution of asphyxia-associated neonatal mortality in São Paulo between 2004 and 2013. The spatial distribution of neonatal mortality in São Paulo between 2006 and 2010 was previously studied by Almeida et al. [22], who demonstrated that the neonatal mortality rate for the 63 microregions of state varied from 6.33 to 16.9/1000 live births. Higher rates were observed in the Northern Coast and in the Southwestern microregions, while lower mortality rates in Northern and Central microregions. In the present study, clusters of asphyxia-associated neonatal mortality in the South, Southeast and Northwest municipalities of the state were identified, demonstrating some overlap of high-risk areas for neonatal mortality between both studies [22]. Differences in risk areas may be explained by different study periods, analytical approaches (municipalities versus microregions) and causes of death (general versus asphyxia-related). It seems relevant that, in both studies, first and second order effects of local spatial dependency and autocorrelation were applied, but only the current study applied a smoothing approach to correct for random spatial fluctuation.

A clear limitation to define neonatal mortality rates in regions or over a time span relies on the fact that these rates may be considered rare events, especially if estimated for microregions, municipalities or cause-specific mortalities. Analytical strategies may be applied to overcome these limitations, such as data aggregation. Almeida et al. [22] applied a spatial aggregation by microregions in addition to time series aggregation. However, precision in defining specific

cities or municipalities within those microregions is compromised. The current study only applied time series aggregation. But even after this procedure, there were 180 municipalities with no asphyxia-associated deaths and 118 municipalities with <500 live births during the study period. Therefore, an analytical dilemma remained: not aggregating spatial data may lead to a significant impact of outlier areas in the spatial distribution, since single cases in small areas (i.e. cities or municipalities) may significantly alter crude rates [23]; while aggregating spatial data may lead to lower precision in higher risk areas. In our study we decided to keep the precision of the information in the municipality level and use time aggregation since temporal evolution of the asphyxia-associated rates was not the objective of this report.

As previously said, neonatal mortality is declining slower than post-neonatal mortality [2], with almost 10% of the deaths being caused by asphyxia/hypoxia [5]. Thus, precisely delimiting areas (i.e. municipalities) with higher risk for cause-specific neonatal mortality may be important for designing public health strategies. Additionally, income is related to neonatal mortality [2, 6–8], and per capita GDP may be a useful indicator for highlighting higher risk areas [8–10], which may be particularly important for middle and low-income countries.

The impact of economic and socioeconomic factors on child mortality has been previously studied. An economic downturn represented by a decrease in per capita GDP, had a negative impact in different child mortality measures, increasing post-neonatal mortality by two fold (95% CI 1.61 to 2.38), child mortality by 2.93 times (95% CI 2.26 to 3.60), and under 5 years of age by 5.44 times (95% CI 4.31 to 6.58) [9]. The impact of economy on child health is potentially caused by reduced access to care, purchasing power and health care resources (number of physicians, hospital beds and staff attending births) [9]. Volpe et al. [8] investigated the association between changes in indicators of health-related resources and variations in infant mortality rates in Brazil from 2000 to 2005. Early neonatal mortality rates were associated with prenatal care (coefficient -0.14 per 1000, $p = 0.026$) and access to sanitation services (-0.04 per 1000, $p = 0.026$). Late neonatal mortality rates were associated with prenatal care (-0.12 per 1000, $p = 0.003$) and rate of cesarean deliveries (0.13 per 1000, $p = 0.005$). Whereas the post-neonatal mortality rate was the only neonatal rate associated with increasing public expenditure on health as a proportion of GDP (-0.76 per 1000, $p = 0.005$), along with prenatal care (-0.64 per 1000, $p<0.001$) and access to water supply (-0.17 per 1000, $p = 0.037$) [8].

Thus, in order to fully explore GIS methods as a supporting tool for improving public health strategies, we intended to both precisely identify cluster areas of cause-specific (asphyxia-associated) neonatal mortality and to explore if areas of lower mean per capita GDP rates were correlated with neonatal mortality. We used a structured and hierarchical spatial analytical approach as an alternative to the dilemma of data aggregation. As previously said, since closer locations (municipalities) tend to share common geographical attributes [11, 12], an empirical Bayesian probabilistic method was applied, which allowed correcting for random fluctuations, especially in locations with small populations and rare events [23]. The local Bayes equation was applied on rates, considering a pondered sum between the crude rates and the local mean value calculated based on the neighbors' rates for each municipality [24]. The rate is inversely proportional to the local population; therefore, it is closer to the crude rate for more populated municipalities [24]. As demonstrated in the current study, the use of the Bayesian approach showed a significant impact on smoothing the asphyxia-associated mortality rates, by helping to identify clusters of higher risk in specific municipalities. Before the smoothing approach, the outliers (areas with small populations) diluted the effect of the clusters. After using the local Bayesian estimates to correct for random spatial fluctuation, we could identify the clusters areas more precisely. Nevertheless, Bayesian estimates for the mean per capita GDP rates did not significantly alter the previously identified clusters, and the maps

before and after the application of the estimates were similar, due to the very essence of per capita GDP rate calculation, which relies on the entire municipality population data.

We identified clusters of high neonatal asphyxia-associated mortality in the South and Southeast, as well as clusters of low mean per capita GDP in similar regions. We found a negative correlation between municipality per capita GDP and asphyxia-associated mortality rates, i.e. low per capita GDP correlated with high neonatal asphyxia-associated mortality municipality rates. Although this correlation was not absolute, it was elevated ($r_s$ = -0.59, 95% CI -0.76 to -0.36), suggesting that the spatial distribution of the neonatal asphyxia-associated mortality can be partially explained by this economic indicator. Certainly, other variables are associated with asphyxia-associated neonatal mortality and should be studied. The correlation between neonatal asphyxia-associated deaths and per capita GDP was exploratory and it was not adjusted for other confounders.

Our study has limitations, including the use of secondary data, which brings an uncertainty to the exact cause of death in the database, as previously described. We intended to overcome this challenge by considering all "asphyxia" terms in any line of the death certificate. The use of aggregated timeseries data, a ten-year period, is also challenging, since, during the study period, the asphyxia-associated neonatal mortality rates varied from 1.37/1000 in 2004 to 0.93/1000 in 2013 live births. It is possible that asphyxia-associated mortality or per capita GDP rates for a specific municipality may have improved or deteriorated during the study period. However, using aggregated data enable us to identify spatial clusters, which would have not been possible if we had considered shorter periods of times.

To our knowledge, this is the first study to demonstrate the spatial distribution of a cause specific neonatal mortality in Brazil. In conclusion, the spatial analysis identified cluster areas (municipalities) of high asphyxia-associated neonatal mortality and low per capita GDP rates, with a significant negative correlation. Spatial cluster identification improved after adjustment for rates' instability using a smoothing approach for rates based on small populations, which ensured more robustness to our findings. The present optimized, structured, and hierarchical approach allowed for the identification of high-risk areas of asphyxia-associated mortality. Therefore, the proposed methodology may be helpful for guiding public health efforts to decrease avoidable causes of neonatal mortality.

## Acknowledgments

We thank all technical staff of SEADE Foundation for their work with the database and Josiane Quintiliano Xavier de Castro, MD, for helping in the deterministic linkage between Live Birth Certificates and Death Certificates. Database use was possible due to agreements #23089.004297/2008-11 and #23089.000057/2014-95 between Fundação SEADE and Universidade Federal de São Paulo.

## Author Contributions

**Conceptualization:** Daniela Testoni Costa-Nobre, Mandira Daripa Kawakami, Ana Sílvia Scavacini Marinonio, Milton Harumi Miyoshi, Tulio Konstantyner, Mônica La Porte Teixeira, Bernadette Waldvogel, Maria Fernanda Branco de Almeida, Ruth Guinsburg, Carlos Roberto Veiga Kiffer.

**Data curation:** Mandira Daripa Kawakami, Kelsy Catherina Nema Areco, Adriana Sanudo, Rosa Maria Vieira Freitas, Liliam Cristina Correia Morais, Mônica La Porte Teixeira, Bernadette Waldvogel.

**Formal analysis:** Daniela Testoni Costa-Nobre, Kelsy Catherina Nema Areco, Carlos Roberto Veiga Kiffer.

**Funding acquisition:** Maria Fernanda Branco de Almeida, Ruth Guinsburg.

**Investigation:** Daniela Testoni Costa-Nobre, Maria Fernanda Branco de Almeida, Ruth Guinsburg, Carlos Roberto Veiga Kiffer.

**Methodology:** Daniela Testoni Costa-Nobre, Adriana Sanudo, Ana Sílvia Scavacini Marinonio, Ruth Guinsburg, Carlos Roberto Veiga Kiffer.

**Resources:** Rosa Maria Vieira Freitas.

**Supervision:** Carlos Roberto Veiga Kiffer.

**Validation:** Paulo Bandiera-Paiva.

**Visualization:** Daniela Testoni Costa-Nobre, Rita Cassia Xavier Balda, Ana Sílvia Scavacini Marinonio, Milton Harumi Miyoshi, Tulio Konstantyner, Liliam Cristina Correia Morais, Bernadette Waldvogel, Maria Fernanda Branco de Almeida, Ruth Guinsburg.

**Writing – original draft:** Daniela Testoni Costa-Nobre.

**Writing – review & editing:** Mandira Daripa Kawakami, Kelsy Catherina Nema Areco, Adriana Sanudo, Rita Cassia Xavier Balda, Ana Sílvia Scavacini Marinonio, Milton Harumi Miyoshi, Tulio Konstantyner, Paulo Bandiera-Paiva, Rosa Maria Vieira Freitas, Liliam Cristina Correia Morais, Mônica La Porte Teixeira, Bernadette Waldvogel, Maria Fernanda Branco de Almeida, Ruth Guinsburg, Carlos Roberto Veiga Kiffer.

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
