## [Decision Letter · Decision Letter 0]

13 May 2021

PONE-D-20-35565

Clusters of cause specific neonatal mortality and its association with per capita gross domestic product: a structured spatial analytical approach

PLOS ONE

Dear Dr. Daniela Testoni Costa-Nobre

Thank you for submitting your manuscript to PLOS ONE. After careful consideration, we feel that it has merit but does not fully meet PLOS ONE’s publication criteria as it currently stands. Therefore, we invite you to submit a revised version of the manuscript that addresses the points raised during the review process.

We look forward to receiving your revised manuscript.

Kind regards,

Wen-Wei Sung, M.D., Ph.D.

Academic Editor

PLOS ONE

Journal Requirements:

2. In the ethics statement in the manuscript and in the online submission form, please provide additional information about the patient records/samples used in your retrospective study, including: a) whether all data were fully anonymized before you accessed them; b) the date range (month and year) during which the data were accessed.

[Fundação de Amparo à Pesquisa de São Paulo-FAPESP, Project # 2017/03748-7]

 [The datasets generated during and/or analysed during the current study are available from the corresponding author on reasonable request.]

5. We note that Figures 2, 3 and 4 in your submission contain map images which may be copyrighted. All PLOS content is published under the Creative Commons Attribution License (CC BY 4.0), which means that the manuscript, images, and Supporting Information files will be freely available online, and any third party is permitted to access, download, copy, distribute, and use these materials in any way, even commercially, with proper attribution. For these reasons, we cannot publish previously copyrighted maps or satellite images created using proprietary data, such as Google software (Google Maps, Street View, and Earth). For more information, see our copyright guidelines: http://journals.plos.org/plosone/s/licenses-and-copyright.

You may seek permission from the original copyright holder of Figures 2, 3 and 4 to publish the content specifically under the CC BY 4.0 license. 

If you are unable to obtain permission from the original copyright holder to publish these figures under the CC BY 4.0 license or if the copyright holder’s requirements are incompatible with the CC BY 4.0 license, please either i) remove the figure or ii) supply a replacement figure that complies with the CC BY 4.0 license. Please check copyright information on all replacement figures and update the figure caption with source information. If applicable, please specify in the figure caption text when a figure is similar but not identical to the original image and is therefore for illustrative purposes only.

Reviewers' comments:

Reviewer's Responses to Questions

**Comments to the Author**

1. Is the manuscript technically sound, and do the data support the conclusions?

Reviewer #1: Yes

Reviewer #2: Partly

Reviewer #3: Partly

2. Has the statistical analysis been performed appropriately and rigorously? 

Reviewer #1: Yes

Reviewer #2: No

Reviewer #3: Yes

3. Have the authors made all data underlying the findings in their manuscript fully available?

Reviewer #1: Yes

Reviewer #2: Yes

Reviewer #3: Yes

4. Is the manuscript presented in an intelligible fashion and written in standard English?

Reviewer #1: Yes

Reviewer #2: Yes

Reviewer #3: Yes

5. Review Comments to the Author

Reviewer #1: The manuscript provides very useful insight into the cause specific neonatal mortality and its association with per capita gross domestic product in upper middle income settings. The methodology of the study is sound and intelligent inferences are made from the statistical analysis. There are a few minor suggestions

Introduction: The Introduction can be further strengthened by clearly setting out the aim of the study highlighting the significance of strengthening the evidence base in this area and why this study is important.

Reviewer #2: The authors conducted a population based study to investigate the cluster areas of asphyxia-associated

neonatal mortality and to explore the per capita gross domestic product (GDP) as an associated risk factor in São Paulo State, Brazil. Ecological study including live births residents in SP from 2004-2013. They concluded that Spatial analysis identified cluster areas of high asphyxia-associated neonatal mortality and low GDP rates, with a significant negative correlation. Overall, the manuscript was well written. I have some comments on the methodology and analysis.

1. One of the major limitations of the study is that there is no adjustment on the other risk factors such as education, population density, income, etc. into their analysis. Is that due to the lack of the information?

2. The authors need to provide a summary table for the demographic factors for the study cohort.

3. The authors conducted spatial moving average maps showed a non-random distribution among municipalities, with presence of clusters (I=0.049; p=0.026). They also conducted local Bayes estimates, and identified much more significant clusters (I=0.587; p=0.001). Why there is big difference between these the results of these two methods?

4. On Page 15, more detailed explanation needs to be provided for the formulae on line 171.

5. What is the significance level of the study? It should be described in the method section.

6. I didn’t find ethnical approval for this study.

Reviewer #3: The study focuses on an interesting and needed topic and the analysis is sounds.

However, there are some relevant limitations. The main one is linked to the statement presented in lines 378-380 [“Therefore, the proposed methodology may be helpful for guiding public health efforts to decrease avoidable causes of neonatal mortality”]. This statement is not fully truth. It is of course extremely useful to assess the distribution of neonatal mortality but the usability of the link with GPD will require further analysis. In fact, other factors could count for this link, as stated by the authors themselves, like health service quality and availability/accessibility and demographic factors (mother age and/or education, for example). None of these data have been considered in order to actually find the factor linked with higher mortality, nor these have been considered in order to adjust the GDP as risk factors. Practically speaking the found relationship is not actually useful to defined which actions to consider reducing neonatal mortality in this areas.

Other key points are the following:

1. The rational for which the study focuses just on asphyxia related neonatal mortality in never stated and unclear. A reason could be that a previous study assessed that link but if that is the case it would be advantageous to clarify this since the beginning instead of just in the discussion. Also, considering the discrepancies between the two studies and the different period it seems even more relevant to explain why avoiding including overall neonatal mortality and possibly, other cause of the same.

2. Please consider adding the proportion of the main causes of neonatal mortality in Brazil (line 79-81)

3. Please consider adding some considerations/hypothesis on why the increase in GDP has only be associated with reduction on post-neonatal mortality in Brazil (line 90-93)

4. Line 127-128: “The mean per capita GDP for the period was calculated for each municipality by dividing the GDP by the estimated population in the same period.” I think it would be clearer adding the bold part “by dividing the GDP of each municipality by the estimated population”, otherwise it looks like that the total SP GDP was divided by people which would imply considering that every person in each municipality has the same level of wealth.

5. Result: please add the total neonatal deaths and the distribution per cause (line 188). Also, data on the number of municipalities with high asphyxia related neonatal deaths rate and no association with GDP [paragraph from line 254] should eb added.

6. PLOS authors have the option to publish the peer review history of their article (what does this mean?). If published, this will include your full peer review and any attached files.

Reviewer #1: No

Reviewer #2: No

Reviewer #3: No

---

## [Author Response · Author response to Decision Letter 0]

16 Jun 2021

We would like to thank the reviewers for their thoughtful suggestions. The responses to the Editor and reviewers are below:

Editor

 We checked the PLOS ONE´s style requirements and follow them. 

2. In the ethics statement in the manuscript and in the online submission form, please provide additional information about the patient records/samples used in your retrospective study, including: a) whether all data were fully anonymized before you accessed them; b) the date range (month and year) during which the data were accessed.

We added a sentence to the manuscript to include the information requested: “The dataset was accessed from October 2018 to June 2021, and it was fully anonymized before the access. 

[Fundação de Amparo à Pesquisa de São Paulo-FAPESP, Project # 2017/03748-7]

 [The datasets generated during and/or analysed during the current study are available from the corresponding author on reasonable request.]

We would like to update our Funding Statement as: Fundação de Amparo à Pesquisa de São Paulo - FAPESP, Project # 2017/03748-7. 

We uploaded the database of the study in the ZENODO public repository as requested: https://zenodo.org/deposit/4915807. The DOI for the data is: 10.5281/zenodo.4915807. The dataset is in closed access right now, but we will change to public access if the paper is accepted. Please update the Data Availability statement on our behalf. 

5. We note that Figures 2, 3 and 4 in your submission contain map images which may be copyrighted. All PLOS content is published under the Creative Commons Attribution License (CC BY 4.0), which means that the manuscript, images, and Supporting Information files will be freely available online, and any third party is permitted to access, download, copy, distribute, and use these materials in any way, even commercially, with proper attribution. For these reasons, we cannot publish previously copyrighted maps or satellite images created using proprietary data, such as Google software (Google Maps, Street View, and Earth). For more information, see our copyright guidelines: http://journals.plos.org/plosone/s/licenses-and-copyright.

 You may seek permission from the original copyright holder of Figures 2, 3 and 4 to publish the content specifically under the CC BY 4.0 license. 

 If you are unable to obtain permission from the original copyright holder to publish these figures under the CC BY 4.0 license or if the copyright holder’s requirements are incompatible with the CC BY 4.0 license, please either i) remove the figure or ii) supply a replacement figure that complies with the CC BY 4.0 license. Please check copyright information on all replacement figures and update the figure caption with source information. If applicable, please specify in the figure caption text when a figure is similar but not identical to the original image and is therefore for illustrative purposes only.

The maps in figures 2, 3 and 4 are not copyrighted. They were created by the research team and all the authors give permission for the open-access journal PLOS ONE to publish them under the Creative Commons Attribution License (CCAL) CC BY 4.0.

Reviewer #1

The manuscript provides very useful insight into the cause specific neonatal mortality and its association with per capita gross domestic product in upper middle-income settings. The methodology of the study is sound and intelligent inferences are made from the statistical analysis. There are a few minor suggestions

Introduction: The Introduction can be further strengthened by clearly setting out the aim of the study highlighting the significance of strengthening the evidence base in this area and why this study is important.

Thank you for your suggestion. We added a sentence at the end of the introduction (before the aim of the study) summarizing the importance of the study: “Since asphyxia/hypoxia is a leading cause of pottentialy preventable neonatal deaths in Brazil, a methodology to identify high risk areas for these deaths and to explore the co-occurrence of associated events would be helpful for guiding public health strategies.”

Reviewer #2

The authors conducted a population based study to investigate the cluster areas of asphyxia-associated neonatal mortality and to explore the per capita gross domestic product (GDP) as an associated risk factor in São Paulo State, Brazil. Ecological study including live births residents in SP from 2004-2013. They concluded that Spatial analysis identified cluster areas of high asphyxia-associated neonatal mortality and low GDP rates, with a significant negative correlation. Overall, the manuscript was well written. I have some comments on the methodology and analysis.

1. One of the major limitations of the study is that there is no adjustment on the other risk factors such as education, population density, income, etc. into their analysis. Is that due to the lack of the information?

Several other factors may impact the distribution of asphyxia-associated neonatal mortality in São Paulo State as stated by the reviewer: education, income, medical training, health care facility distribution, quality of prenatal care, among others. The aim of our study was not to analyze risk factors for asphyxia-associated neonatal deaths, but to identify the cluster areas where these deaths occur and to explore if the methodology of spatial analysis is helpful to identify risk factors associated to the geographic distribution of asphyxia-associated neonatal death. In this context, we described the aim of our study as “to investigate by a structured and hierarchical spatial analytical method the occurrence of clustering areas of asphyxia-associated neonatal mortality and to explore the mean per capita GDP as an associated risk factor…” Future studies need to be done in order to identify additional risk factors. We reinforced this idea at the end of the discussion section adding the sentence: “Certainly other risk factors are associated with asphyxia-associated neonatal mortality and should be studied. The correlation between neonatal asphyxia-associated deaths and per capita GDP was exploratory and it was not adjusted for other risk factors.”

2. The authors need to provide a summary table for the demographic factors for the study cohort.

We included the demographic characteristics for the study cohort on the first paragraph of the results section: “The mean birth weight of the neonates that died with asphyxia was 1,907g (±SD 1,154g); with 17 infants with unknown birth weight. More than half were male (56% [3752/6713]. Regarding gestational age, 37% [2,484/6,642] had 37 to 41 weeks, 15% [978/6,642] 32 to 36 weeks, 16% [1,036/6,642] 28 to 31 weeks, 32% [2,090/6,642] 22 to 27 weeks, and 1% [54/6,642] ≥42 weeks, with 71 infants with unknown gestational age. Maternal age was <20 years for 21% [1,391/6,713], 20-34 years for 65% [4,361/6,713], and > 34 years for 14% [961/6,713].”

3. The authors conducted spatial moving average maps showed a non-random distribution among municipalities, with presence of clusters (I=0.049; p=0.026). They also conducted local Bayes estimates, and identified much more significant clusters (I=0.587; p=0.001). Why there is big difference between these the results of these two methods?

The difference found shows the importance of the methodology. Before the smoothing approach, the outliers (areas with small populations) diluted the effect of the clusters. After using the local Bayes estimates to correct for random spatial fluctuation, we could identify the clusters areas more precisely. The previous sentence was included in the discussion section. 

4. On Page 15, more detailed explanation needs to be provided for the formulae on line 171.

We added more information explaining the formulae for the local empiric Bayesian estimator equation: “The larger the population in area I, the closer weight wi, which means that in areas where the population is larger, the empirical Bayes estimate is very close to the risk rate for that area (ti), and in areas with a small population, the value of bi will be intermediary between ti and mi.”

5. What is the significance level of the study? It should be described in the method section.

“A p-value <0.05 was considered as statistically significant for all analyses.” We added this sentence at the end of the methods section. 

6. I didn’t find ethnical approval for this study.

Thank you for your note. We added the ethical approval for the study after the first paragraph in the methods section: “The study was approved by the Ethics Committee on Human Research of Escola Paulista de Medicina – Universidade Federal de São Paulo (#2,580,929) and by the Board of Directors of Fundação SEADE”.

Reviewer #3

 The study focuses on an interesting and needed topic and the analysis is sounds.

However, there are some relevant limitations. The main one is linked to the statement presented in lines 378-380 [“Therefore, the proposed methodology may be helpful for guiding public health efforts to decrease avoidable causes of neonatal mortality”]. This statement is not fully truth. It is of course extremely useful to assess the distribution of neonatal mortality but the usability of the link with GPD will require further analysis. In fact, other factors could count for this link, as stated by the authors themselves, like health service quality and availability/accessibility and demographic factors (mother age and/or education, for example). None of these data have been considered in order to actually find the factor linked with higher mortality, nor these have been considered in order to adjust the GDP as risk factors. Practically speaking the found relationship is not actually useful to defined which actions to consider reducing neonatal mortality in these areas.

Several other factors may impact the distribution of asphyxia-associated neonatal mortality in São Paulo State as stated by the reviewer: education, income, medical training, health care facility distribution, quality of prenatal care, among others. The aim of our study was not to analyze risk factors for asphyxia-associated neonatal deaths, but to identify the cluster areas where these deaths occur and to explore if the methodology of spatial analysis is helpful to identify risk factors associated to the geographic distribution of asphyxia-associated neonatal deaths. Future studies need to be done in order to identify the complex interaction of multiple risk factors associated with neonatal deaths in the clusters identified in this study. We reinforce the limitation of the association found between asphyxia-associated mortality and per capita GDP adding the following sentence at the end of the results section: “Certainly other risk factors are associated with asphyxia-associated neonatal mortality and should be studied. The correlation between neonatal asphyxia-associated deaths and per capita GDP was exploratory and it was not adjusted for other risk factors.”

Other key points are the following:

1. The rational for which the study focuses just on asphyxia related neonatal mortality in never stated and unclear. A reason could be that a previous study assessed that link but if that is the case it would be advantageous to clarify this since the beginning instead of just in the discussion. Also, considering the discrepancies between the two studies and the different period it seems even more relevant to explain why avoiding including overall neonatal mortality and possibly, other cause of the same.

We focused our study on the asphyxia-associated neonatal mortality because asphyxia/hypoxia is a leading cause of potentially preventable neonatal deaths in Brazil. We added a sentence in the introduction to highlight the importance of our study: “Since asphyxia/hypoxia is a leading cause of pottentialy preventable neonatal deaths in Brazil, a methodology to identify high risk areas for these deaths and to explore the co-occurrence of associated events would be helpful for guiding public health strategies.” Also, several other studies describe the overall neonatal mortality in Brazil, but there are few, if any, reporting cause-specific neonatal mortality. 

2. Please consider adding the proportion of the main causes of neonatal mortality in Brazil (line 79-81)

We added that information as suggested: “Despite its considerable decline over the years, neonatal mortality remains high, with almost 10% of the underlying cause of deaths being classified as asphyxia/hypoxia. Asphyxia/hypoxia is the 5th cause of neonatal mortality in Brazil following prematurity (30%), congenital malformation (23%), infections (19%) and maternal factors, but it represents a pottentialy preventable one” 

3. Please consider adding some considerations/hypothesis on why the increase in GDP has only be associated with reduction on post-neonatal mortality in Brazil (line 90-93)

We added the hypothesis given by the authors of the cited study: “For Volpe et al, the increase in public expenditure on health reduced the post-neonatal mortality with no impact on neonatal mortality because the strategies for reducing neonatal mortality are more expensive than those for reducing post-neonatal mortality, for which low-cost public interventions may have a great effect”. 

4. Line 127-128: “The mean per capita GDP for the period was calculated for each municipality by dividing the GDP by the estimated population in the same period.” I think it would be clearer adding the bold part “by dividing the GDP of each municipality by the estimated population”, otherwise it looks like that the total SP GDP was divided by people which would imply considering that every person in each municipality has the same level of wealth.

Thanks for the suggestion. We changed the sentence to: “The mean per capita GDP for the period was calculated for each municipality by dividing the GDP of each municipality by the estimated population in the same period.”

5. Result: please add the total neonatal deaths and the distribution per cause (line 188). Also, data on the number of municipalities with high asphyxia related neonatal deaths rate and no association with GDP [paragraph from line 254] should eb added.

The total number of neonatal deaths was 50,993. We added this information to Figure 1. Unfortunately, we cannot add the distribution per cause because we just have the database for deaths associated with asphyxia. The study dataset will be available in the ZENODO public respiratory: https://zenodo.org/deposit/4915807. 

There were 63 municipalities with high asphyxia-associated neonatal deaths and no significance (by LISA) for mean per capita GDP. We added a sentence with this information at the end of the results section: “There were 63 municipalities in clusters of high asphyxia-associated neonatal deaths but no significance (by LISA) for mean per capita GDP rates”.

---

## [Decision Letter · Decision Letter 1]

9 Jul 2021

PONE-D-20-35565R1

Clusters of cause specific neonatal mortality and its association with per capita gross domestic product: a structured spatial analytical approach

PLOS ONE

Dear Dr. Daniela Testoni Costa-Nobre,

Thank you for submitting your manuscript to PLOS ONE. After careful consideration, we feel that it has merit but does not fully meet PLOS ONE’s publication criteria as it currently stands. Therefore, we invite you to submit a revised version of the manuscript that addresses the points raised during the review process.

We look forward to receiving your revised manuscript.

Kind regards,

Wen-Wei Sung, M.D., Ph.D.

Academic Editor

PLOS ONE

Reviewers' comments:

Reviewer's Responses to Questions

**Comments to the Author**

1. If the authors have adequately addressed your comments raised in a previous round of review and you feel that this manuscript is now acceptable for publication, you may indicate that here to bypass the “Comments to the Author” section, enter your conflict of interest statement in the “Confidential to Editor” section, and submit your "Accept" recommendation.

Reviewer #1: All comments have been addressed

Reviewer #2: All comments have been addressed

Reviewer #3: (No Response)

2. Is the manuscript technically sound, and do the data support the conclusions?

Reviewer #1: Yes

Reviewer #2: Yes

Reviewer #3: Partly

3. Has the statistical analysis been performed appropriately and rigorously? 

Reviewer #1: Yes

Reviewer #2: Yes

Reviewer #3: N/A

4. Have the authors made all data underlying the findings in their manuscript fully available?

Reviewer #1: Yes

Reviewer #2: Yes

Reviewer #3: Yes

5. Is the manuscript presented in an intelligible fashion and written in standard English?

Reviewer #1: Yes

Reviewer #2: Yes

Reviewer #3: Yes

6. Review Comments to the Author

Reviewer #1: Thank you for successfully addressing the points highlighted in the previous detailed review. The manuscript is now suitable for publication in the journal.

Reviewer #2: (No Response)

Reviewer #3: Thank you for the replies to the previous questions and the edits to the paper.

However, it is clear even from the authors replies that what has been explored is the association between one specific cause of death and GDP. This association is defined as a “risk factor” but this has not be proven. Only the investigation of others listed risk factors could have clarified whether the risk factor is the low GDP, or, for example, the low access to care, low education, etc. It is sure that low GDP is likely to be associated with such risk factors but “having and association” and “being a risk factors” are two different issues.

The authors themselves stated that “The aim of our study was not to analyse risk factors for asphyxia-associated neonatal, but to identify the cluster areas where these deaths occur and to explore if the methodology of spatial analysis is helpful to identify risk factors associated to the geographic distribution of asphyxia-associated neonatal death". Therefore, the manuscript could be considered if the objective is focus in the geospatial distribution but avoiding the definition of GDP as a risk factor (from objective and discussion).

7. PLOS authors have the option to publish the peer review history of their article (what does this mean?). If published, this will include your full peer review and any attached files.

Reviewer #1: **Yes: **Dr Syed M Abbas

Reviewer #2: No

Reviewer #3: No

---

## [Author Response · Author response to Decision Letter 1]

13 Jul 2021

Response for Reviewer #1 and 2

Thank you for your inputs in the previous revised manuscript. 

Response for Reviewer #3: 

Thank you for the replies to the previous questions and the edits to the paper.

However, it is clear even from the authors replies that what has been explored is the association between one specific cause of death and GDP. This association is defined as a “risk factor” but this has not be proven. Only the investigation of others listed risk factors could have clarified whether the risk factor is the low GDP, or, for example, the low access to care, low education, etc. It is sure that low GDP is likely to be associated with such risk factors but “having and association” and “being a risk factors” are two different issues. The authors themselves stated that “The aim of our study was not to analyze risk factors for asphyxia-associated neonatal, but to identify the cluster areas where these deaths occur and to explore if the methodology of spatial analysis is helpful to identify risk factors associated to the geographic distribution of asphyxia-associated neonatal death". Therefore, the manuscript could be considered if the objective is focus in the geospatial distribution but avoiding the definition of GDP as a risk factor (from objective and discussion).

We completely agree with you. We cannot say that we have studied a risk factor for asphyxia-associated neonatal mortality given the methodology applied. Our objective was to investigate by a spatial analytical approach the occurrence of clustering areas of asphyxia-associated neonatal deaths and to explore the mean per capita GDP distribution in these areas. You are right that we should be more careful with our language in order to avoid misunderstandings. We have reviewed the manuscript to exclude or modify all the sentences that could give the idea that we were studying a risk factor instead of exploring a possible association between per capita GDP and asphyxia-associated neonatal deaths:

• In the objective of the abstract, we changed the following sentence “To investigate the cluster areas of asphyxia-associated neonatal mortality and to explore the per capita gross domestic product (GDP) as an associated risk factor in São Paulo State (SP), Brazil” to “To investigate the cluster areas of asphyxia-associated neonatal mortality and to explore its association with per capita gross domestic product (GDP)” (Revised manuscript with tracked changes – lines 45-47).

• In the last sentence of the 4th paragraph of the Introduction section, we changed the sentence from “Spatial analysis may be used to explore the co-occurrence of a health event and other population events on the same geographic unit (i.e. economic, educational, sociological indicators) as possible risk factors” to “Spatial analysis may be used to explore the co-occurrence of a health event and other population events on the same geographic unit (i.e. economic, educational, sociological indicators)” (Revised manuscript with tracked changes – lines 104-106).

• We changed the objective at the end of introduction section from “our aim was to investigate by a structured and hierarchical spatial analytical method the occurrence of clustering areas of asphyxia-associated neonatal mortality and to explore the mean per capita GDP as an associated risk factor” to “our aim was to investigate by a structured and hierarchical spatial analytical method the occurrence of clustering areas of asphyxia-associated neonatal mortality and to explore its association with mean per capita GDP” (Revised manuscript with tracked changes – lines 110-112).

• In the 8th paragraph of the discussion section, we changed the following: “Certainly, other risk factors are associated with asphyxia-associated neonatal mortality and should be studied. The correlation between neonatal asphyxia-associated deaths and per capita GDP was exploratory and it was not adjusted for other risk factors”; to “Certainly, other variables are associated with asphyxia-associated neonatal mortality and should be studied. The correlation between neonatal asphyxia-associated deaths and per capita GDP was exploratory and it was not adjusted for other confounders” (Revised manuscript with tracked changes – lines 392-395).

• In the last paragraph of the discussion section, we changed “The present optimized, structured, and hierarchical approach allowed for the identification of high-risk areas of asphyxia-associated mortality and an associated risk factor (GDP)” to “The present optimized, structured, and hierarchical approach allowed for the identification of high-risk areas of asphyxia-associated mortality.” (Revised manuscript with tracked changes – lines 410-411)

We thank you for your thoughtful suggestions to improve our manuscript.

---

## [Decision Letter · Decision Letter 2]

27 Jul 2021

Clusters of cause specific neonatal mortality and its association with per capita gross domestic product: a structured spatial analytical approach

PONE-D-20-35565R2

Dear Dr. Daniela Testoni Costa-Nobre,

We’re pleased to inform you that your manuscript has been judged scientifically suitable for publication and will be formally accepted for publication once it meets all outstanding technical requirements.

Kind regards,

Wen-Wei Sung, M.D., Ph.D.

Academic Editor

PLOS ONE

Reviewers' comments:

Reviewer's Responses to Questions

**Comments to the Author**

1. If the authors have adequately addressed your comments raised in a previous round of review and you feel that this manuscript is now acceptable for publication, you may indicate that here to bypass the “Comments to the Author” section, enter your conflict of interest statement in the “Confidential to Editor” section, and submit your "Accept" recommendation.

Reviewer #1: All comments have been addressed

Reviewer #2: All comments have been addressed

Reviewer #3: All comments have been addressed

2. Is the manuscript technically sound, and do the data support the conclusions?

Reviewer #1: Yes

Reviewer #2: Yes

Reviewer #3: Yes

3. Has the statistical analysis been performed appropriately and rigorously? 

Reviewer #1: Yes

Reviewer #2: Yes

Reviewer #3: Yes

4. Have the authors made all data underlying the findings in their manuscript fully available?

Reviewer #1: Yes

Reviewer #2: Yes

Reviewer #3: Yes

5. Is the manuscript presented in an intelligible fashion and written in standard English?

Reviewer #1: Yes

Reviewer #2: Yes

Reviewer #3: Yes

6. Review Comments to the Author

Reviewer #1: The authors have successfully addressed the points highlighted in the previous review. The manuscript is now suitable for publication in the journal.

Reviewer #2: (No Response)

Reviewer #3: Thank you for editing the paper. All comments have been addressed. I have n further comments to add.

7. PLOS authors have the option to publish the peer review history of their article (what does this mean?). If published, this will include your full peer review and any attached files.

Reviewer #1: No

Reviewer #2: No

Reviewer #3: No

---

## [Editor Report · Acceptance letter]

9 Aug 2021

PONE-D-20-35565R2 

Clusters of cause specific neonatal mortality and its association with per capita gross domestic product: a structured spatial analytical approach 

Dear Dr. Testoni Costa-Nobre:

I'm pleased to inform you that your manuscript has been deemed suitable for publication in PLOS ONE. Congratulations! Your manuscript is now with our production department. 

Kind regards, 

on behalf of

Dr. Wen-Wei Sung 

Academic Editor

PLOS ONE